# Molecular Dynamics and Evolution of Centromeres in the Genus Equus

**DOI:** 10.3390/ijms23084183

**Published:** 2022-04-10

**Authors:** Francesca M. Piras, Eleonora Cappelletti, Marco Santagostino, Solomon G. Nergadze, Elena Giulotto, Elena Raimondi

**Affiliations:** Department of Biology and Biotechnology “L. Spallanzani”, University of Pavia, 27100 Pavia, Italy; mfrancesca.piras@unipv.it (F.M.P.); eleonora.cappelletti@unipv.it (E.C.); marco.santagostino@unipv.it (M.S.); solomon.nergadze@unipv.it (S.G.N.); elena.raimondi@unipv.it (E.R.)

**Keywords:** satellite DNA, neocentromeres, centromere repositioning, centromere sliding, karyotype evolution, CENP-A, genus Equus

## Abstract

The centromere is the chromosomal locus essential for proper chromosome segregation. While the centromeric function is well conserved and epigenetically specified, centromeric DNA sequences are typically composed of satellite DNA and represent the most rapidly evolving sequences in eukaryotic genomes. The presence of satellite sequences at centromeres hampered the comprehensive molecular analysis of these enigmatic loci. The discovery of functional centromeres completely devoid of satellite repetitions and fixed in some animal and plant species represented a turning point in centromere biology, definitively proving the epigenetic nature of the centromere. The first satellite-free centromere, fixed in a vertebrate species, was discovered in the horse. Later, an extraordinary number of satellite-free neocentromeres had been discovered in other species of the genus Equus, which remains the only mammalian genus with numerous satellite-free centromeres described thus far. These neocentromeres arose recently during evolution and are caught in a stage of incomplete maturation. Their presence made the equids a unique model for investigating, at molecular level, the minimal requirements for centromere seeding and evolution. This model system provided new insights on how centromeres are established and transmitted to the progeny and on the role of satellite DNA in different aspects of centromere biology.

## 1. Introduction

The centromere is the chromosomal locus driving chromosome segregation via kinetochore formation and spindle attachment. As a consequence, a functional centromere is required for the stable inheritance of each chromosome during mitosis and meiosis.

The simplest centromere found in nature is the “point centromere” of the budding yeast *Saccharomyces cerevisiae*, where a single modified nucleosome is associated to an about 120 bp long DNA sequence that is both necessary and sufficient to grant full centromere function [1]. An opposite situation is observed in some animal and plant species where “holocentromeres” are spread along the length of the entire chromosomes, the best studied example being the one of *Caenorhabditis elegans* [2]. However, most centromeres of higher eukaryotes, including humans, are of the regional type [1]. These centromeres occupy a defined region along the chromosome, typically encompassing megabases of highly repetitive DNA, called satellite DNA [3]. This type of sequences occupies the functional centromeric domains and the surrounding pericentromeric regions.

*S. cerevisiae* is the only eukaryotic organism in which the centromeric function is entirely determined by the DNA sequence [4,5]. In all other eukaryotes, although the centromeric function is well conserved, rapid evolution among and within species has been observed for centromeric DNA sequences and size, making centromeres rapidly evolving genomic regions that can shape karyotypes and drive speciation. The conflict between the need of function conservation and the surprising variety of DNA sequences and strategies involved in centromere formation is known as the “centromere paradox” [6]. A possible interpretation of this paradox is that epigenetic determinants are contributing to the maintenance and inheritance of the centromeric function [5,7,8]. A large body of evidence supports this hypothesis. To this regard, an important observation is the occasional stable transmission of dicentric chromosomes following inactivation of one centromere [9]. Therefore, epigenetic mechanisms define whether a DNA sequence can acquire a centromeric function. Another piece of evidence supporting the epigenetic basis for the centromeric function comes from human clinical neo-centromeres, which confer the ability to segregate correctly to acentric fragments thanks to the acquisition of centromeric function by a hitherto non-centromeric interstitial sequence [10,11]. The discovery of naturally occurring satellite-free centromeres fixed in some animal and plant species indicates that satellite DNA is not strictly required for the centromeric function. These centromeres coexist, in the same karyotype, with canonical satellite-based centromeres and are perfectly functional. The first satellite-free centromere, fixed in a vertebrate species, was discovered in one horse chromosome by our group [12]. We demonstrated that this chromosome is totally devoid of satellite repeats both within the CENP-A binding domain and in the surrounding pericentromeric regions. Thereafter, other examples of satellite-free centromeres have been described in other vertebrate species by our laboratory [13,14] and other groups [15,16,17] as well as in plants [18,19,20,21,22]. The major epigenetic mark of regional centromeres is CENP-A, a centromere-specific histone H3 variant which interacts with a number of proteins constitutionally or transiently associated to centromeric DNA to establish the kinetochore [23]. Overexpression of CENP-A in Drosophila causes its misincorporation in non-centromeric regions and promotes the formation of functional ectopic kinetochores, thus inducing chromosome segregation errors [24,25]; a similar situation was observed in budding yeast [26]. These experiments strongly suggest that CENP-A is both necessary and sufficient for the formation of the kinetochore and that the presence of this histone variant drives the formation of a highly specialized chromatin structure that directs the formation of a functional centromere. Similar experiments of CENP-A overexpression and mistargeting to non-centromeric regions in human chromosomes failed to induce complete kinetochore assembly, suggesting that this process is more complex and additional functions are required [27,28]. Nevertheless, mislocalisation of CENP-A alters the structure and function of endogenous kinetochores and leads to chromosomal instability, suggesting a key role of this protein in specifying centromere function [29].

## 2. Centromeric and Pericentromeric Satellite DNA

Satellite DNA (SatDNA) is composed by extended tandemly repeated monomers typically located at centromeres but, in some species, also at terminal and intrachromosomal sites [30,31,32]. Satellite DNA arrays show wide variation in copy number and sequence composition among species, representing the most rapidly evolving sequences in eukaryotic genomes and contributing to shape their karyotypes [31]. SatDNA tends to be homogenized following the principles of concerted evolution. Homogenization of these repeats may occur by unequal crossing over, gene conversion, rolling circle replication or transposition [33,34]. Unequal exchanges and mutational variation in individual satellite monomers may cause the formation of higher-order repeats (HORs) consisting of large units composed by multiple divergent monomers [35,36]. Different families of satellite DNA can be located at the centromere functional core and in the flanking pericentromeric regions [37].

The presence of satellite sequences in the majority of eukaryotic centromeres suggests that they may play a role in kinetochore assembly and/or stability, as inferred since their discovery [38,39,40]. Even if their sequences are highly divergent, centromeric satellites from different species tend to be AT-rich [41] and are composed of monomers of similar length (often around 150–180 bp), leading to hypothesise that the repeat unit itself might reflect uniformity in nucleosome phasing and that these constraints, rather than the sequence itself, are needed for heterochromatin propagation [6,42,43,44]. However, it is difficult to reconcile the existence of centromeres completely devoid of satellite DNA with a pivotal function of this genome component.

The function of satellite DNA remains poorly understood. Pericentromeric and centromeric satellite DNA families are transcribed in a number of animal and plant species (examples are reported in [45,46,47,48,49,50,51,52,53]; reviewed in [54,55]). Although pericentromeric satellites are embedded in heterochromatin, several findings indicate that pericentric heterochromatin is unexpectedly permissive to transcription. The resulting pericentric transcripts were proposed to contribute to the maintenance of the heterochromatic condensed state (reviewed in [30,56]). Transcription has also been reported for centromeric satellites, and centromeric transcripts were proposed to be involved in CENP-A loading and kinetochore assembly [53,57,58]. However, it is still a matter of debate whether transcription is required for proper centromeric function. Indeed, transcription is not a universal feature of centromeres, with satellite-free centromeres arising either in transcribed regions [11,59,60,61,62,63] or in gene desert regions without evidence of transcription [11,12,59,60,64].

Although dispensable for centromere specification, it was suggested that satellite DNA might contribute to the stability of centromeres, as predicted by the “centromere drive” theory [65,66]. According to this model, during the asymmetric female meiosis, homologous chromosomes may compete for their inclusion in the egg via their “centromere strength”. Since centromeres with more extended satellite DNA arrays may host more CENP-A nucleosomes, it was suggested that centromere strength is related to the length of tandemly repeated sequences [65,66,67,68,69]. Recent cell biological studies indicate that centromeres compete by destabilizing microtubule interactions [70,71]. At the same time, an excessive accumulation of repeated arrays by “selfish” centromeres would lead to expansion of the CENP-A binding domain, which seems to be counteracted by the co-evolution between centromeric proteins and centromeric DNA [65]. In this scenario, satellites at the borders of the centromeric core, during expansion, become more and more degenerated, preventing centromeric protein binding. These degenerate repeats occupy pericentromeric regions, lose their transmission advantage and evolve neutrally [66].

The extreme abundance of pericentromeric compared to centromeric satellite DNA seems in contradiction with the common view that pericentromeric satellites are simply selfish parasitic DNA sequences [72], or fossils of centromere evolution [3]. The maintenance of such extended arrays of tandemly repeated DNA would be too high a burden for the cell [73]. To solve this controversy, pericentromeric satellites were recently proposed to have a structural role in the tridimensional nuclear organisation, driving the formation of chromocenters [73].

The typical presence of satellite repeats at centromeres hampered a comprehensive molecular analysis of these intriguing loci. The discovery of functional centromeres completely devoid of satellite repeats represented a milestone in centromere biology [7,10,11,12,13,15,16,17,74].

## 3. Evolutionarily New Centromeres

The term “centromerization” was coined by Choo to define the process of centromere formation in a chromosomal region [7]. Centromerization normally concerns the propagation of an existing centromere during replication. Rarely, this phenomenon occurs in regions which are normally non-centromeric. The ectopic centromere that appears occasionally in a hitherto non-centromeric chromosomal region is called neocentromere [7,8,74]. Two different types of neocentromeres have been identified: human clinical neocentromeres and evolutionarily new centromeres (ENC).

While the majority of human neocentromeres are seeded in acentric fragments, conferring on them the ability to segregate, a few human neocentromeres have been described in otherwise normal karyotypes [10,75]. In these cases, the old centromere appears unchanged, but functionally inactive; conversely, a non-centromeric sequence acquires the ability to recruit a functional kinetochore.

While clinical human neocentromeres are sporadic cases that are not fixed in the population, evolutionarily new centromeres are fixed in some species and represent an aspect of karyotype evolution. Interestingly, several lines of evidence suggest that these two types of neocentromeres are faces of the same coin [76], since the chromosome regions where some human neocentromeres were seeded were orthologous to evolutionary neocentromeres in primates [64,77,78,79]. These observations indicate that there are chromosome domains which have an inherent potentiality to form centromeres, suggesting that a similar mechanism may originate both human neocentromeres and ENCs [79]. In other words, some chromosome domains may harbor the capability to form centromeres depending on still unknown sequence features or on a peculiar chromatin environment which may favor centromere seeding.

Evolutionarily new centromeres are generated during evolution by repositioning events. Centromere repositioning consists of the movement of the centromere along the chromosome length in the absence of any structural rearrangement. In other words, while the ancestral centromere is inactivated, a centromere appears de novo at another chromosomal position. The first unequivocal proof of the existence of this phenomenon was given by Montefalcone and colleagues in 1999 [80]. The group traced the phylogenetic history of chromosome IX in primates by comparative FISH with a panel of BAC clones. They demonstrated that, in some species, the centromere moved and was repositioned along the chromosome while the surrounding markers maintained the original position. Thereafter, a number of ENCs were found in primates and in other mammalian orders (reviewed in [76,81]).

The ENCs initially described in primates are composed of the classical extended arrays of satellite DNA. The first example of a satellite-free evolutionary neocentromere was described in the horse by our laboratory [12]. Later on, our group discovered that the species of the genus Equus (horses, asses and zebras) are characterized by an extraordinary number of satellite-free neocentromeres [13,14]. Beyond equids, the only satellite-free centromere identified so far in a mammalian species was discovered in orangutan chromosome 12 [17]. Differently from the equids, this satellite-free centromere and the ancestral satellite-based centromere coexist in the same population as a polymorphism [17]. Thus, the presence of extraordinary numbers of satellite-free centromeres has made the equids a unique model for dissecting several aspects of centromere biology [82,83,84].

## 4. Unique Model System: The Genus Equus

The genus Equus is the only extant genus of the Equidae family, which belongs to the Perissodactyla (odd-toed ungulates) order together with Tapiridae (tapirs) and Rhinocerotidae (rhinoceroses) families. Currently living equids comprise horses (*Equus caballus* and *Equus przewalskii*), African asses (*Equus africanus asinus* and *Equus africanus somaliensis*), Asiatic asses (*Equus hemionus* and *Equus kiang*), and zebras *(Equus grevyi*, *Equus burchelli*, and *Equus zebra hartmannae*).

While the karyotypes of Tapiridae and Rhinocerotidae remained quite stable during evolution and resemble the putative perissodactyl ancestral karyotype, characterized by high chromosomal number and a prevalence of acrocentric chromosomes [85], Equus karyotypes underwent a rapid evolution after the divergence from the common ancestor, dated 4.0–4.5 million years ago (Mya) [85,86]. The most recent radiation events, found among asses and zebras, occurred less than 1 Mya and many species and subspecies emerged in this very short evolutionary time [85,87,88].

The evolutionary plasticity of the equid genomes is also proved by the high mobility of insertion DNA elements, such as retrotransposons [89], interstitial telomeres [90,91] and numts (nuclear sequences of mitochondrial origin) [92]. The insertion of these sequences is recognized as a driving force in evolution [91,92,93,94], and the extraordinary frequency of insertion polymorphism in the genus Equus [89,90,92] supports the hypothesis that the genome of equid species is in an ongoing evolutionary state.

The numerous speciation events that occurred during Equus’ evolution were accompanied by extensive karyotype reshaping due to both chromosome rearrangements and centromere repositioning [14,88,95,96,97]. These phenomena were responsible for a reduction in chromosome number and for the passage from an ancestral karyotype, with the majority of chromosomes being acrocentric, to karyotypes mainly composed by meta- and submeta-centric chromosomes. Several of these (sub)metacentrics harbour a repositioned centromere and, in the majority of them, this centromere is satellite-free [14,96]. The chromosomal distribution of satellite tandem repeats, analysed by FISH in *Equus caballus*, *E. asinus*, *E. grevyi* and *E. burchelli* [14], revealed that several centromeres are devoid of satellite DNA (Figure 1) and that blocks of satellite DNA sequences are present at several non-centromeric chromosome end (Figure 1B–D). The presence of these large satellite DNA arrays at non-centromeric terminal positions on a subset of (sub)metacentrics with a repositioned satellite-less centromere, combined with the knowledge that ancestral chromosomes were known to be acrocentric, strongly suggests that they are relics of ancestral centromeres.

These observations led us to propose a model explaining the birth and evolution of new centromeres [14]. According to this model, satellite-free centromeres are the primary result of a centromere repositioning event which moved the centromeric function from the original position to a new one lacking satellite DNA. On the other hand, the satellite sequences of the old inactivated centromere remain at their original position and are then progressively lost during evolution. Meanwhile, “immature” satellite-free centromeres progressively reach their maturity by acquiring satellite DNA. Thanks to their rapid evolution, the equids offered the possibility to recognize all the steps of this process (Figure 1), while only “mature” neocentromeres, associated with satellite DNA, were initially described in primates [76].

The extraordinary number of satellite-free centromeres in equids is a direct consequence of their rapid and recent evolution [14,88,96,97]. The number of satellite-free centromeres is surprisingly high in asses and zebras (Figure 1), whose species radiated in very recent evolutionary times. Moreover, only in these species were satellite DNA sequences at non-centromeric ends identified as fossil remains of the ancient inactivated centromeres, testifying that these repositioning events were extremely recent. On the other hand, in the horse, whose karyotype shows a prevalence of acrocentric chromosomes and is considered more similar to the ancestral perissodactyl karyotype, only one satellite-free neocentromere is present, suggesting that this species is more “mature” with respect to centromere evolution.

Satellite-free natural centromeres fixed in the Equus species [12,13,14] offer the opportunity to investigate the minimal requirements for centromere seeding, evolution and complete maturation at the molecular level. In particular, using this model system, several key open questions can be addressed: How are the centromeric domains organized at the molecular level? How are centromeric domains transmitted through generations? Which mechanisms drive centromere birth and maturation during evolution? Which is the contribution of satellite DNA to segregation fidelity and inhibition of recombination during meiosis?

### 4.1. Birth, Evolution and Transmission of Satellite-Free Centromeres

The discovery that the centromere of horse chromosome 11 (ECA11) is completely devoid of satellite DNA [12] was the result of ChIP-on-chip experiments carried out hybridizing horse genomic DNA, purified from chromatin immunoprecipitated with an anti CENP-A antibody, to an array spanning the region containing the primary constriction. The satellite-free centromeric locus of ECA11 does not contain segmental duplications nor protein coding genes. This region is highly conserved in mammals, but the only species in which a centromere is present therein is the horse. These findings are in agreement with the hypothesis that the centromeric function is unrelated to DNA sequence and that the ECA11 centromere was formed very recently during the evolution of the horse lineage. This centromere, in spite of being functional and stable in all horses, did not acquire all the marks typical of mammalian centromeres. It represents the first example of an evolutionary neocentromere caught in a stage of incomplete maturation.

In these early experiments, two peaks corresponding to two CENP-A binding domains were identified in the individual analysed. The same domains were identified when chromatin was immunoprecipitated with an anti CENP-C antibody. When the same experiment was then carried out in five unrelated individuals, it was demonstrated that each horse exhibited a distinct arrangement of CENP-A binding domains [98]. Two CENP-A binding regions were present in three of the five individuals and, in the other two individuals, only one broad peak was observed, which was the result of partial overlapping of two peaks. While single peaks occupy about 100 kb, the entire ECA11 region involved occupies about 500 kb. Single-nucleotide polymorphism (SNP) analysis gave the definitive proof that, when two peaks were observed in the same individual, they were not located on the same chromosome, but each homolog contained a single CENP-A binding domain. These different positional alleles for CENP-A binding were defined as “epialleles” and the phenomenon was called “centromere sliding”. Positional variation in centromeric domains was also reported within the satellite-based centromere of human chromosome 17, where the centromere can assemble on different alpha satellite arrays [99]. Further evidence of the existence of epialleles was derived by single molecule analysis by immuno-FISH on chromatin fibres. Taken together, these results demonstrated that the position of the CENP-A binding domain can slide within a relatively wide region and allowed us to conclude that centromeric domains are characterized by positional instability which may be physically limited by epigenetic boundaries.

Centromeric domains were then analysed in the donkey by means of ChIP-seq [13], demonstrating that, in this species, more than half of the centromeres (16 out of 31) are devoid of satellite DNA. The 16 satellite-free donkey centromeric domains derived from centromere repositioning events that occurred in this lineage since they are orthologous to horse non-centromeric sequences. Considering that horse and donkey lineages separated recently (about 4–4.5 Mya), we can speculate that there was not enough evolutionary time for satellite DNA accumulation and centromere maturation.

The donkey satellite-free centromeres spanned 54–345 kb and, as already described for the horse satellite-free centromere, contained one or two CENP-A binding domains. The sequence of these domains was de novo assembled and analysed, demonstrating that all of them lay in LINE- and AT-rich regions. AT richness is a typical feature of centromeres in a number of organisms [4,11,100] and might promote the adoption of a non-B DNA configuration which is typically found at centromeres [36,101]. It was recently shown that AT-rich exogenous DNA is also capable of functioning as a centromere in the model organism *Schizosaccharomyces pombe* [102]. It was proposed that AT-rich sequences, which display low nucleosome occupancy, may favour the insertion of LINE-1 elements, which are frequently enriched in natural human centromeres [34] and in clinical neocentromeres [11,79,103]. A peculiar sequence organisation was described in Drosophila, where centromeric domains form on islands enriched in LINE retroelements which are embedded within extended arrays of pericentromeric satellite repeats [104]. It is not clear whether AT and LINE richness contributes to the acquisition of centromeric function. Although these sequences are enriched in the donkey satellite-free centromeres, their abundance is not observed at the evolutionary satellite-free neocentromere of horse chromosome 11 [12,13]. The observation that LINE/LTR-rich domains cluster within the nucleus suggests that their abundance at several centromeric domains may be related to function [105]. Therefore, the sequence composition of the satellite-free donkey centromeres may be involved in the organisation into subnuclear domains that promote the functional activation of centromeric chromatin.

The sequences of the donkey satellite-free centromeres were compared with their horse non-centromeric counterparts demonstrating, in some instances, the presence of rearrangements (deletions, amplifications, insertions and inversions). These rearrangements could have occurred before or after neocentromere formation. As it was suggested that CENP-A can be recruited to DNA breaks [106], chromosome rearrangements may have promoted centromere formation. However, this is not a rule, since, in several instances, no relevant sequence change was found in the donkey with respect to the horse. Similarly, Tolomeo and colleagues demonstrated that the sequence underlying the satellite-free polymorphic ENC that they described in the orangutan is identical to its non-centromeric counterpart [17].

Five donkey neocentromeres contain novel tandem repetitions of chromosome-specific sequences that are a single copy in the horse genome. These amplified genomic sequences are unrelated to one another, with amplified units raging in size from a few to several tens of kilobases. The repeat copy number was variable in the two individuals analysed, suggesting the existence of polymorphism in the population. It is tempting to hypothesize that DNA amplification may represent an intermediate stage toward satellite DNA formation during evolution [13]. In light of these observations, a model for the maturation of a centromere during evolution was proposed, including different routes, some of which involve sequence amplification (Figure 2). Following this new model, after centromere inactivation, associated satellite sequences are maintained at the site of the ancestral centromere while a neocentromere arises in a new locus, completely devoid of satellite sequences (Figure 2B). Subsequently, satellite sequences are gradually lost at the non-centromeric site (Figure 2C), and, at the functional satellite-free centromere, amplification may occur (Figure 2E,F) as an intermediate step toward complete maturation of the neocentromere. The amplifications observed at some donkey centromeres may be considered as an early seed of chromosome-specific centromeric satellites, suggesting that amplification-like mechanisms can trigger the formation of tandemly repeated DNA sequences within the centromere core. Finally, these neocentromeres will acquire the typical complexity of the mammalian centromeres and will be embedded in tandemly repeated DNA (Figure 2D). It cannot be excluded that sequence amplification may occur before neocentromere formation (Figure 2G) and may favour centromerization, but no data supporting this alternative route are available. It is important to underline that, as indicated in Figure 2, horse or donkey chromosomes corresponding to each maturation step were observed, thus representing evolutionary snapshots which validate the proposed model.

The possibility of interspecific breeding in Equus species offered the opportunity to follow the transmission of epialleles for CENP-A binding through generations and, thus, to understand how centromeric domains are transmitted to the progeny [13]. Since the hybrid individuals (mules and hinnies) contain two haploid genomes, one from the donkey and one from the horse, the transmission of individual epialleles across generations could be followed. The analysis of CENP-A binding domains in hybrid families revealed that they are inherited as Mendelian traits. However, their position can slide in one generation. Interestingly, the centromeric region in the offspring is always at least partially overlapping the domain of the parent, suggesting that CENP-A nucleosomes are partially maintained at their position. Therefore, centromeres do not jump to a completely new location but, rather, in the course of several generations, slight movements may accumulate, giving rise to nonoverlapping epialleles. Conversely, the position of the centromere is stable during mitotic propagation of cultured cells that were previously immortalized [107], suggesting that the sliding that was observed in the hybrids could have occurred during germ-line differentiation, meiotic division, fertilization or early developmental stages.

### 4.2. Satellite DNA and Centromere Stability

The presence of well-characterized satellite-free centromeres in a context of canonical satellite-based centromeres in the equid model system was exploited to understand the contribution of satellite DNA to different aspects of centromere biology.

Beyond the extraordinary number of evolutionary neocentromeres, the high plasticity of the equid genomes is also testified by the architectural organisation of centromeric and pericentromeric satellite DNA families. Three main satellite DNA families were described, namely 37cen, 2PI and 137sat. These satellite families differ in the length of their repeat unit, with 37cen consisting of a 221 bp repeat [14,108], 2PI composed of a 23 bp repeat [14] and 137sat made by 137 bp repetitions [109].

While in asses and zebras, the majority of centromeres are satellite-free and many satellite loci are not centromeric, the horse displays a unique satellite-free centromere coexisting with the typical satellite-based mammalian centromeres. High-resolution FISH on combed DNA fibres demonstrated that at least some horse satellite-based centromeres may display a mosaic arrangement of the different satellite DNA families where short arrays of the 2PI and EC137 satellites are closely intermingled and immerged within very large stretches of the 37cen sequence [109]. This organisation suggests that recombination events among centromeric and pericentromeric satellite DNA can occur in the horse genome. The arrays of 37cen embed the centromeric core of horse satellite-based centromeres. Indeed, immunoprecipitation experiments with an anti-CENP-A antibody proved that this satellite family is the only one bound by CENP-A and thus bears the centromeric function [52]. Although centromeric satellites are typically AT-rich [36], 37cen is GC-rich indicating that GC richness is compatible with the centromeric function. 

The organisation of horse satellite-based centromeres, with a centromeric core of 37cen satellite and a flanking pericentromere characterized by different sequence composition, is in agreement with the typical organisation of mammalian centromeres. Indeed, it is known that new satellite sequences arise and expand in the inner centromeric core, progressively moving the older satellite families towards the pericentromere, forming layers of different age [110]. Pericentromeric satellites progressively degenerate, losing the ability to be bound by centromeric proteins and avoiding a harmful expansion of the functional centromere [65,66].

As mentioned above, centromeric satellite DNA is transcribed in several species. It has been hypothesized that transcriptional competence may allow centromeric chromatin to assume a partially open conformation that may be important for CENP-A loading. Centromere transcripts have been proposed to favour the formation of a flexible scaffold needed to assembly or stabilize the kinetochore [30,111]. Moreover, they may act in trans on all or on a subset of chromosomes independently of the primary DNA sequence [30,111]. In agreement with this notion, the horse 37cen centromeric satellite was proved, by RNA-seq approach, to be transcriptionally active [52]. However, the possible role of this centromeric transcript in the epigenetic establishment of centromeric chromatin is still under debate.

Satellite DNA is believed to stabilize centromeres, and a crucial aspect of centromere biology concerns its contribution to chromosome segregation fidelity. Data on the mitotic stability of satellite-free centromeres were already obtained by the analysis of pathologic satellite-free human centromeres. These neocentromeres are often present in the individual in mosaic form, indicating mitotic instability. However, most human pathologic neocentromeres give rise to partial trisomy or tetrasomy and, therefore, the selective disadvantage of cell lines carrying partial aneuploidy rather than an intrinsic mitotic instability of the neocentromere itself may be responsible for the mosaicism observed [11].

The coexistence of canonical satellite-based centromeres with the satellite-free centromere of chromosome 11 was exploited to test whether satellite DNA may influence chromosome segregation in a non-pathologic setting. The mitotic stability of this neocentromere was investigated and compared with that of horse chromosome 13, which is similar in size but has a canonical satellite-based centromere [112]. To this purpose, two independent molecular-cytogenetic approaches, the micronucleus assay and interphase aneuploidy analysis, were used. The mitotic stability of the two chromosomes was the same demonstrating that, in the horse system, centromeric satellite DNA is dispensable for chromosome segregation fidelity. Although the widespread presence of repeated DNA at natural centromeres suggests that there is a positive selection for this arrangement, the above data demonstrate that the mitotic stability of a chromosome is not universally influenced by the presence of highly repeated DNA sequences at its centromere.

### 4.3. Meiotic Behaviour of Centromeric Domains

An important aspect of centromere biology is the so-called “centromere effect”, that is the suppression of meiotic recombination exerted by the centromere [113]. This phenomenon has been described in all eukaryotes, including humans and other mammals, but it was a matter of debate whether the presence of repetitive DNA contributes to it [114].

The horse model appeared again as the right system to answer this question [115]. Using cytogenetic mapping of recombination foci with respect to centromeres in horse pachytene spreads, it was discovered that the horse neocentromere exerts a crossover suppression as well as canonical centromeres. Thus, this inhibitory effect depends on the centromeric function itself rather than on the presence of highly repetitive DNA [115]. This discovery reinforces the hypothesis that this effect is related to the epigenetic environment of the centromere and is in agreement with the notion that the occurrence of crossovers near centromeres is selectively disadvantageous because it may cause premature sister chromatid separation leading to non-disjunction at the second meiotic division [114,115]. It would be interesting to verify whether centromeres that were inactivated during evolution would have acquired the recombination potential typical of interstitial loci.

During the analysis of the distribution of recombination foci in horse meiosis, a peculiar phenomenon was observed: while the majority of meiotic bivalent chromosomes was labelled with a single immunofluorescence centromeric signal, double-spotted or extended centromeric signals were also detected on a subset of autosomal bivalents with a satellite-based centromere [115]. It is well known that the centromeres of the homologous chromosomes are paired in the chromosome meiotic bivalent [116,117], but the centromeric domains of the paired homologous chromosomes appeared shifted with respect to each other. While the same phenomenon was detected, but remained unexplained, in a few mammalian species affecting very few chromosomes [118,119], the horse showed a surprisingly high frequency of these “double-spotted” centromeres, and their number was variable from cell to cell within the same individual [115]. Their appearance was proposed to be the result of two components: positional variation of the CENP-A binding domains between the two homologous chromosomes and misalignment between the centromeric and pericentromeric satellite DNA arrays during pairing [115]. According to this model, the positional variation of CENP-A binding domains, which was observed at the molecular level at satellite-free centromeres, seems to also affect satellite-based centromeres. Thus, the centromeric domains in the paired homologous chromosomes seem to occupy different positions along the 37cen array. In addition, the polymorphism in the number of centromeric tandem repeats combined with the high sequence identity among monomers may result in a staggered pairing of the homologous satellite arrays. According to this interpretation, misalignment is variable from meiosis to meiosis and may increase the physical distance between centromeric domains, which may already be in different positions on the two homologs, making centromere sliding visible at the cytogenetic resolution [115]. Clearly, this could not happen at satellite-free centromeres where no repeats are present, and thus the centromere sliding phenomenon could not be “magnified” to be visible at cytogenetic resolution.

## 5. Conclusions

One of the key enigmas of chromosome biology is how centromeric chromatin is established, transmitted to the progeny and modified during evolution. This is one of the most challenging open questions in biology due to its implications in human cancer and chromosome abnormalities as well as in karyotype evolution and speciation.

Centromeres are typically associated with satellite DNA sequences which hinder the molecular analysis of centromeric chromatin. The discovery of a natural centromere completely devoid of satellite DNA and fixed in a species, the horse, represented a milestone in centromere biology. The subsequent identification of satellite-free centromeres in other equids as well as in other vertebrate and plant species has revealed that this type of centromeric DNA configuration is not so rare in natural karyotypes. However, the genus Equus remained unprecedented in terms of uncoupling between satellite DNA and centromeric function. This is the only mammalian genus in which an extraordinary number of satellite-free centromeres were identified so far. Thus, the genus Equus represents a powerful model system in centromere biology. The main results obtained using this system demonstrated that: (1) CENP-A binding domains are positionally unstable, generating epialleles which are inherited as mendelian traits; (2) amplified DNA sequences are possible intermediates towards centromere maturation during evolution; (3) the majority of satellite-free centromeres are enriched in AT and LINE sequences; (4) satellite DNA is dispensable for chromosome segregation fidelity and (5) the centromere effect is not related to the presence of satellite DNA but to the centromeric function itself.

## Figures and Tables

**Figure 1 ijms-23-04183-f001:**
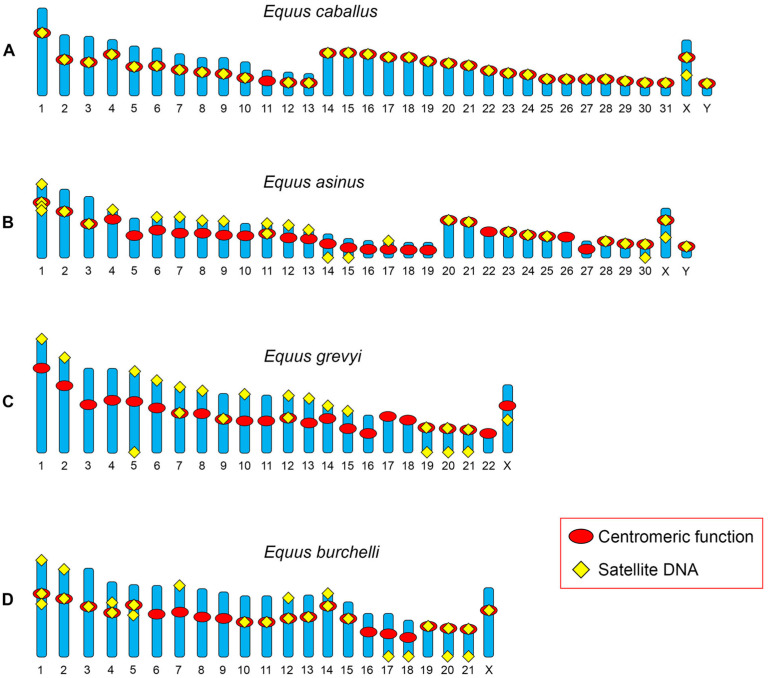
Schematic representation of the chromosomal distribution of satellite DNA obtained by FISH experiments in *E. caballus*, *E. asinus*, *E. grevyi* and *E. burchelli*. (**A**) In *E. caballus*, all centromeres, with the exception of that of chromosome 11, are associated with satellite DNA sequences. Satellite DNA is also present at an interstitial position on the long arm of chromosome X. (**B**) In *E. asinus*, 18 centromeres are not associated with satellite DNA sequences at the FISH resolution level. Satellite DNA sequences are present at 13 non-centromeric chromosome ends. Satellite DNA is present at an interstitial position on the long arm of chromosome X. (**C**) In *E. grevyi*, 17 centromeres are void of satellite DNA and satellite DNA sequences are present at 15 non-centromeric termini and at an interstitial position of the long arm of the X chromosome. (**D**) In *E. burchelli*, 7 centromeres do not display satellite DNA sequences and 9 non-centromeric chromosome ends are associated with satellite DNA. Satellite DNA loci are present in the pericentromeric region of chromosomes 1, 4 and 5. Modified from [14].

**Figure 2 ijms-23-04183-f002:**
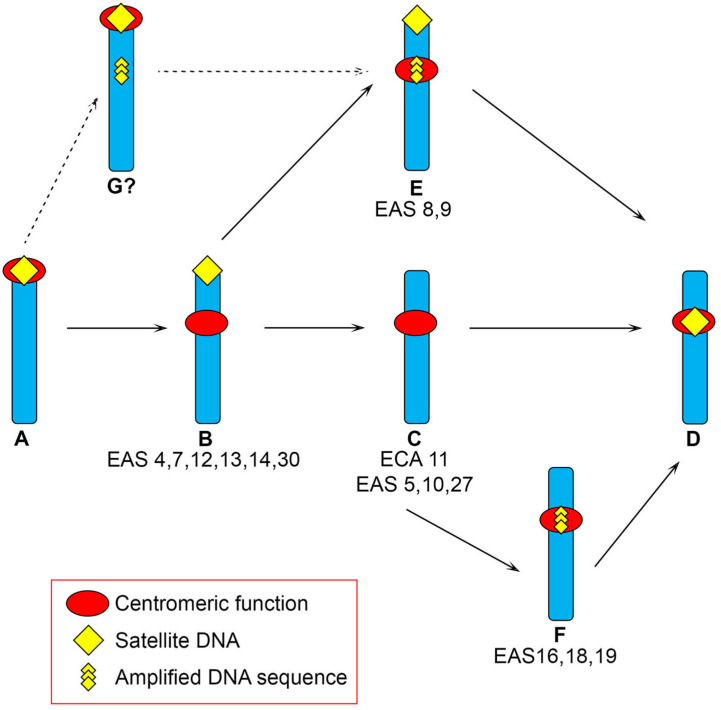
Model for the maturation of a centromere during evolution. Different routes are delineated leading to a mature satellite-based repositioned centromere (**D**) from an ancestral satellite-based centromere (**A**) through satellite-free intermediates (**B**,**C**,**E**,**F**). Route A–D: a neocentromere arises in a satellite-free region and satellite DNA remains at the ancestral position; the ancestral satellite DNA is lost from the non-centromeric terminus and, finally, the neocentromere acquire satellite repeats, giving rise to a “mature” neocentromere. This route follows the previously proposed model [14]. Routes A, B, E, D or A, B, C, F, D: at an already functional satellite-free neocentromere, amplification occurs as an intermediate step toward complete maturation of the neocentromere. Neocentromere maturation and loss of satellite DNA from the old centromere site are independent events that can occur at different stages. The 16 donkey chromosomes carrying a satellite-free neocentromere and the horse chromosome 11 exemplify satellite-free intermediates (**B**,**C**,**E**,**F**) and are listed below each step. It cannot be excluded that sequence amplification precedes neocentromere formation (**G**), but no chromosome corresponding to this step was found so far. Modified from [13,14].

## Data Availability

Not applicable.

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
