# Peer review of "Molecular Dynamics and Evolution of Centromeres in the Genus Equus"

_ijms, 2022, doi:10.3390/ijms23084183_

Round 1
Reviewer 1 Report
This review is mostly well-written and comprehensive. My main comment is that the authors should be more careful in the Introduction when they discuss epigenetic versus genetic contributions to centromeres. They first state that there is “conclusive proof demonstrating that the centromere is uncoupled from the underlying DNA sequence” (lines 57-58). The evidence suggests that specific DNA sequences are not required for centromere function, but does not support such a strong statement that sequence makes no contribution to function. The “conclusive proof” sentence also seems inconsistent with later statements: “satellite DNA might contribute to the stability of centromeres” (lines 114-115) and “satellite DNA is believed to stabilize centromeres and a crucial aspect of centromere biology concerns its contribution to chromosome segregation fidelity” (lines 420-421). Furthermore, the centromere drive hypothesis depends on a genetic contribution to centromere function, which is also inconsistent with the “conclusive proof” sentence. The authors should decide which side they want to take (epigenetic vs genetic centromeres) and maintain consistency.
It’s also not clear how the centromere paradox is “solved” by maintaining centromeres epigenetically (lines 48-50). Rather, the proposed centromere drive solution to the paradox depends on centromere DNA sequence contributing to centromere function (i.e., not completely epigenetic).
Additional comments:
CENP-A is not “ubiquitous” (line 65), as insect lineages don’t appear to have CENP-A at all (see PMID 25247700, 34813757).
The statement that centromere strength is “defined as the ability of their kinetochores to recruit microtubules” (lines 117-118) is misleading. The recent cell biological studies (as opposed to theoretical models) indicate that centromeres compete by destabilizing microtubule interactions, not by binding more microtubules (see PMID 29097549, 31402175). Additionally, it’s not clear what the authors mean by centromere strength being “directly proportional to the length of the arrays of satellite DNA” (lines 118-119). The authors should explain what “directly proportional” means and how they arrived at this conclusion.
The statement that “an excessive accumulation of repeated arrays by selfish centromeres would lead to a harmful expansion of the centromeric core” (lines 119-120) is unclear. What is the centromeric core, and what is the evidence that expansion is harmful?
The idea that “there are chromosome domains which have an inherent potentiality to form centromeres” (lines 155-156) is interesting. Can the authors speculate about how this potential might be explained by DNA sequence or some other property (such as chromatin state) that is conserved between primate species?
Some concluding thoughts would be beneficial, as the review ends abruptly.
Minor notes for grammar and clarity:
147-149: probably should be “inactive; conversely”
214: "led to" -> "led us to"
236: "at molecular level" -> "at the molecular level"
266: "example of evolutionary" -> "example of an evolutionary"
282: "alfa" -> "alpha"
283: "epialleles derived by" -> "epialleles was derived by"
285: "allowed to conclude" -> "allowed us to conclude"
290-292: "The ... centromeric domains ... are not centromeric"
This sentence is difficult to understand and should be reworded.
315: "non centromeric" -> "non-centromeric"
420: probably should be “centromeres, and”
Reviewer 2 Report
Centromeres in the genus Equus are of great interest due to their unique evolutionary plasticity. These centromeres are characterized by very high instability, there is an unprecedented high level of evolutionary neocentromeres, many centromeres are devoid of the classical structures that accompany centromeres - satellites and LINE elements. The manuscript is very interesting and relevant. It is well and clearly written: very concise, yet very capacious. Almost all questions that arise in the course of reading the text are answered in the following text.
What questions seemed to me not spelled out clearly enough:
Satellite DNA is found both in the centromeres themselves (zones of chromosomes marked with CENP A) and in the pericentromeric heterochromatin. Despite the fact that the review has a separate chapter with the heading “2. Centromeric and pericentromeric satellite DNA”, the text of the manuscript constantly raises the question of whether the authors mean centromeric or pericentromere + centromere chromatin when they say “satellite-free centromeres”.
This seems to me very important, because the main function of the centromeric chromatin is to maintain an epigenetically stable inheritance of CENP A, but the most important functions of the pericentromeric chromatin is to create a stable contact of sister chromatids during metaphase, which is ensured by the preservation of cohesin in the pericentromeric regions until the onset of anaphase. It is the loss of this function that makes karyotypes with a small amount of pericentromeric heterochromatin very unstable. It seems to me that the authors should prescribe more clearly what is meant by non-satellite centromeres throughout the text. For example, here: “These centromeres occupy a defined region along the chromosome, typically encompassing megabases of highly repetitive DNA, called satellite DNA [3].”.
The absence of satellites in the centromeres themselves, along with the presence of a huge number of them in the pericentromeric heterochromatin, has recently been shown in Drosophila. For many years, it was believed that centromeres in Drosophila lie in satellite DNA blocks; however, it has recently been shown that CENP A binds predominantly to islands of LINE class elements; moreover, a transposon characterizing all centromeres has been identified (Chang et al., 2019, PlosBiol// Islands of retroelements are major components of Drosophila centromeres). .
I was most surprised that some centromeres in the group described by the authors lack LINE elements. It seems to me that this moment requires more emphasis, since it is LINE elements that characterize many centromeres, moreover, it has been shown that they can participate in their functioning. For example, in neocentromeres analysed in (Chueh et al., PLoS Genetics 2009), nockdown of the transcribed LINE element completely blocked neocentromere activity.
The article only briefly touches on an issue that I think is very important. It is a question of suppression of recombination in the vicinity of centromeres and the disappearance of this suppression in the "former" centromeres. It seems to me that the suppression or new acquisition of recombination is very important mechanism governing the evolution of centromeres.
The last remark: the final conclusion section is missing in the paper. In the current form, the manuscript seems incomplete.
Thus, I believe that the article deserves to be published in the IJMS journal after a minor revision.
